

# Inferring neutral winds in the ionospheric transition region from AGW-TID observations with the EISCAT VHF radar and the Nordic Meteor Radar Cluster

Florian Günzkofer[1], Dimitry Pokhotelov[1], Gunter Stober[2], Ingrid Mann[3], Sharon L. Vadas[4], Erich Becker[4], Anders Tjulin[5], Alexander Kozlovsky[6], Masaki Tsutsumi[7,8], Njål Gulbrandsen[9], Satonori Nozawa[10], Mark Lester[11], Evgenia Belova[12], Johan Kero[12], Nicholas J. Mitchell[13,14], and Claudia Borries[1]

[1]Institute for Solar-Terrestrial Physics, German Aerospace Center (DLR), Neustrelitz, Germany
[2]Institute of Applied Physics & Oeschger Center for Climate Change Research, Microwave Physics, University of Bern, Bern, Switzerland
[3]Institute of Physics and Technology, UiT, Arctic University of Norway, Tromsø, Norway
[4]North West Research Associates (NWRA), Boulder, Colorado, USA
[5]EISCAT Scientific Association, Kiruna, Sweden
[6]Sodankylä Geophysical Observatory, University of Oulu, Finland
[7]National Institute of Polar Research, Tachikawa, Japan
[8]The Graduate University for Advanced Studies (SOKENDAI), Tokyo, Japan
[9]Tromsø Geophysical Observatory, UiT - The Arctic University of Norway, Tromsø, Norway
[10]Institute for Space-Earth Environmental Research, Nagoya University, Nagoya, Japan
[11]University of Leicester, Leicester, UK
[12]Swedish Institute of Space Physics (IRF), Kiruna, Sweden
[13]British Antarctic Survey, Cambridge, UK
[14]University of Bath, Bath, UK

**Correspondence:** Florian Günzkofer (florian.guenzkofer@dlr.de)

**Abstract.** Atmospheric Gravity Waves and Traveling Ionospheric Disturbances can be observed in the neutral atmosphere and the ionosphere at a wide range of spatial and temporal scales. Especially at medium scales, these oscillations are often not resolved in general circulation models and are parameterized. We show that ionospheric disturbances forced by upward propagating atmospheric gravity waves can be simultaneously observed with the EISCAT Very High Frequency incoherent scatter radar and the Nordic Meteor Radar Cluster. From combined multi-static measurements, both vertical and horizontal wave parameters can be determined by applying a specially developed Fourier filter analysis method. This method is demonstrated using the example of a strongly pronounced wave mode that occurred during the EISCAT experiment on 7 July 2020. Leveraging the developed technique, we show that the wave characteristics of Traveling Ionospheric Disturbances are notably impacted by the fall transition of the mesosphere/lower thermosphere. We also demonstrate the application of using the determined wave parameters to infer the thermospheric neutral wind velocities. Applying the dissipative anelastic gravity wave dispersion relation, we obtain vertical wind profiles in the lower thermosphere.



## 1 Introduction

Waves balanced by gravity and buoyancy forces are often referred to as gravity waves and originate in various fluids (Andrews, 2010). In the Earth's atmosphere, atmospheric gravity waves (AGWs) can be observed at a wide range of altitudes from the troposphere well into the thermosphere. In the ionosphere, AGWs can be observed as Medium-Scale Traveling Ionospheric Disturbances (MS-TIDs) from neutral-ion collisions (Nicolls et al., 2014). Typical MS-TID wave periods are approximately $15 - 80$ min (Kirchengast et al., 1996; Hocke and Schlegel, 1996). In this region, the wave is subject to electromagnetic effects (Kelly, 2009) in addition to the buoyancy and gravity forces and viscous damping (Pitteway and Hines, 1963; Vadas, 2007). The wavelengths and periods of AGW-TIDs depend on the generation mechanism and the state of the background atmosphere. These disturbances can be forced either by ionospheric processes (Brekke, 1979) or by upward propagating gravity waves generated in the lower or middle atmosphere (Bauer, 1958; Hung et al., 1978; Vadas and Crowley, 2010; Smith, 2012; Nishioka et al., 2013; Azeem et al., 2015, 2017; Frissell et al., 2016; Xu et al., 2019; Becker et al., 2022a) or in the thermosphere via multi-step vertical coupling (Vadas and Becker, 2019; Becker et al., 2022a; Vadas et al., 2023; Vadas et al., 2023). Since neutral-ion collisions create TIDs from AGWs if a component of the AGWs velocity vector lies along the Earth's magnetic field line (e.g., Nicolls et al., 2014), we refer to these waves as AGW-TIDs independent of their generation region or mechanism in this study.

The wave picture in the thermosphere/ionosphere can be highly complicated with several wave modes present. AGW-TIDs forced in the lower atmosphere are capable of propagating to these altitudes only under certain atmospheric conditions. This wave filtering (Lindzen, 1981; Smith, 1996) has a major impact on the mesosphere/lower thermosphere (MLT) region (Holton, 1992; Smith, 2012; Becker, 2012). Consequently, a large number of studies have investigated this impact (see e.g., Hoffmann et al., 2010, 2011; Ern et al., 2011; Smith et al., 2017; Sarkhel et al., 2022, and references therein). Strong changes in AGW-TID activity in the MLT are caused by the seasonal variation of mesospheric mean winds (Stober et al., 2021d), especially during the spring and fall equinox transitions. The latter in particular has been shown to impact tidal waves in the mesosphere (Stober et al., 2020; Pedatella et al., 2021) and possibly well up in the thermosphere (Günzkofer et al., 2022). Investigating the impact of the MLT fall transition, the change of the MLT wind system around the autumn equinox (Stober et al., 2021d), on AGW-TIDs is one of the central topics of this manuscript.

The impact of background winds on AGW-TIDs can be seen from the wave dispersion relations, derived for zero-viscosity (Hines, 1960), for small viscosity in a steady state solution (Pitteway and Hines, 1963), and for full viscosity for wave packets in an anelastic formulation (Vadas and Fritts, 2005). These dispersion relations show that horizontal and vertical wave characteristics are strongly dependent on the neutral atmosphere parameters and dynamics. Considering that the AGW-TID parameters can be derived from MLT observations, thermospheric neutral winds along the AGW propagation direction can be deduced by making use of the above-mentioned dispersion relations (Vadas and Nicolls, 2009). Since neutral wind velocities are difficult to measure at altitudes $\gtrsim 100$ km (Mitchell and Beldon, 2009), this would provide valuable information on thermosphere dynamics.

However, simultaneous measurements with sufficient vertical resolution and horizontal coverage to determine gravity wave



parameters are difficult to obtain, in particular, the required spatial coverage to derive the horizontal wave numbers is often not available. One possibility to perform such three-dimensional measurements is the use of phased array radars. This has been demonstrated for both incoherent scatter radars (ISR) (e.g., Nicolls and Heinselman, 2007; Vadas and Nicolls, 2008, 2009) and coherent scatter radars (Rapp et al., 2011; Stober et al., 2013, 2018). Under certain assumptions, similar measurements are also possible using a classical ISR with multi-beam or beam-swinging capabilities (Nicolls et al., 2014). Other studies applied simultaneous measurements of a single-beam ISR and a GNSS receiver network to extract vertical/horizontal wavelength (van de Kamp et al., 2014). Applying high-resolution GNSS measurements of the total electron content (TEC) to detect and study MS-TIDs is a well-established method (Saito et al., 1998; Tsugawa et al., 2007; Onishi et al., 2009). However, since TEC is the height-integrated electron density, multiple wave modes might be mixed, which makes vertical and horizontal measurements of single wave modes more challenging and prone to observational biases.

In this work, a new strategy is presented utilizing measurements from the EISCAT ISR and the Nordic Meteor Radar Cluster (Stober et al., 2021c, 2022). Thereby, upward-propagating gravity waves can be observed simultaneously with the horizontally resolved wind fields obtained from the meteor radar measurements and ISR measurements that have a high vertical resolution. Since the Nordic Meteor Radar Cluster allows altitude-resolved measurements, it is possible to obtain vertical and horizontal wavelength, wave period, and propagation direction of an individual wave mode. Horizontal wavelengths can be assumed to be constant for a horizontally constant background wind field (Lighthill, 1978; Vadas and Nicolls, 2008). This is usually the case above the turbopause and has been confirmed in measurements (Nicolls and Heinselman, 2007). The further structure of the paper is as follows:

Section 2 will give an overview of the instruments utilized and the specifics of the analyzed measurements. The process of separating different wave modes and determining the wave parameters is presented in Section 3. The method we use to infer neutral winds via the gravity wave dispersion relations will be demonstrated there as well. Section 4 will present AGW-TID measurements conducted during the EISCAT campaign of autumn 2022, both before and after the MLT fall transition. This illustrates the impact of atmospheric transitions on ionospheric dynamics. In Section 5, the advantages and disadvantages of combined EISCAT and Nordic Meteor Radar Cluster AGW-TID measurements compared to previous approaches are discussed. The results are compared to these previous studies in Section 5 as well and possibilities for future work are discussed. Our conclusions are given in Section 6.

## 2 Instruments

### 2.1 EISCAT Incoherent Scatter Radar

A general overview of the different EISCAT radars and experiments can be found in Tjulin (2021). We summarize the information on apparatuses and modes applied to this work.

The EISCAT Scientific Association operates a Very High Frequency (VHF) ISR with a frequency of 224 MHz near Tromsø, Norway (69.6° N, 19.2° E) (Folkestad et al., 1983). The VHF transmitter is operated at a maximum power of about 1.5 MW and the co-located receiver antenna consists of four rectangular (30 m x 40 m) dishes.



All measurements were conducted in the *manda zenith common program 6 (CP6)* mode with the transmitter and co-located
receiver pointed at 90° elevation. This mode allows measurements up to $\sim 200$ km altitude with a high vertical resolution
ranging from several hundred meters in the lower thermosphere to about 10 km at the highest altitudes. The plasma parame-
ters have been obtained with version 9.2 of the Grand Unified Incoherent Scatter Design and Analysis Package (GUISDAP)
(Lehtinen and Huuskonen, 1996). The time resolution of the obtained plasma parameters is determined by the post-experiment
integration time of the ISR raw data which has been set to 60 s. Data from two EISCAT measurement campaigns are utilized
for this study:

**Summer 2020**

The first campaign was conducted on three consecutive days in July 2020 (7th to 9th). The EISCAT VHF radar was operated
from 0 - 12 UTC on each of these days. One advantage of this campaign is the continued low geomagnetic activity during
the three days of the observation campaign with Kp $< 2$ on all three days. The reduced ionospheric variability due to external
forcing provides more favorable conditions for the detection of AGW-TIDs originating in the lower atmosphere. TID detection
is done manually with a coherent wave structure being present for at least two wave periods and exhibiting downward phase
progression. The relative electron density variations should be approximately $\delta N_e / N_e \sim 0.01 - 0.05$ (Vadas and Nicolls, 2009;
Nicolls et al., 2014). On 7 July 2020, a pronounced TID signature was found. This TID was used as a reference to implement
and optimize the applied analysis method to isolate and separate different wave modes and to determine the wave parameters.

**Autumn 2022**

The second campaign was conducted during Autumn 2022 in two separate measurement intervals on 1 September and 13
October. This ensures that measurements are available before and after the MLT fall transition which is expected to take place
over several days around the autumn equinox (Stober et al., 2021d). On both days, the EISCAT VHF radar was operated from
8 - 13 UTC using the same experiment mode as during the summer campaign. This period was chosen to minimize the impact
of geomagnetic substorms which hamper the detection of TIDs. However, both measurement days did indicate the presence
of TIDs. The highest geomagnetic activity during these two measurements occurred on 1 September around 9 UTC with
Kp $= 2.333$.

## 2.2 Nordic Meteor Radar Cluster

Meteor radars have proven to be valuable and reliable instruments to measure neutral winds in the MLT region. These winds
contain valuable information about atmospheric waves such as gravity waves, tides, and planetary waves at the MLT (e.g., Fritts
et al., 2010; de Wit et al., 2016, 2017; McCormack et al., 2017; Pokhotelov et al., 2018; Stober et al., 2021b, d). The wind
velocity is determined by measuring the Doppler shift of the coherent radar scattering from the thermalized plasma generated
by meteoroids entering the Earth's atmosphere and forming an ambipolar diffusing plasma trail (Herlofson, 1951; Greenhow,
1952; McKinley, 1961; Poulter and Baggaley, 1977; Jones and Jones, 1990; Hocking et al., 2001; Stober et al., 2021a). For





this work, we analyze measurements from the high-resolution 3DVAR+DIV retrieval, which is a part of the ASGARD (Agile Software for Gravity wAve Regional Dynamics), of the Nordic Meteor Radar Cluster (Stober et al., 2021c, 2022). This Cluster consists of four meteor radars located in Tromsø (Norway; 69.6° N, 19.2° E), Alta (Norway; 70.0° N, 23.3° E), Kiruna (Sweden; 67.9° N, 21.1° E) and Sodankylä (Finland; 67.4° N, 26.6° E). The retrieved 3D wind fields cover the Nordic countries from $\sim 66°$ N $- 72°$ N latitude and $\sim 12.5°$ E $- 31.5°$ E longitude. The horizontal grid resolution is 30 km and wind

measurements are available from $\sim 80 - 100$ km altitude at 2 km vertical resolution and 10 min timesteps. This higher temporal resolution is possible due to the multi-static measurements that result in a much higher meteor trail detection rate within the overlapping observation volume compared to a monostatic meteor radar. For the visualization of horizontally resolved measurements and the correct geographic mapping with minimal projection errors, we leverage the *m_map* software package (Pawlowicz, 2020).

## 3  Methods

The analysis of AGW-TIDs requires the processing of the ISR and meteor radar measurements. The applied techniques to extract and separate different wave modes from the radar data are described in this section using the data collected on 7 July 2020. We then leverage this methodology and apply the same procedures to observations carried out during several campaigns.

### 3.1  EISCAT

Previous studies suggested that the magnetic field-aligned ion velocity is the most promising ISR parameter to detect TIDs (Williams, 1989; Vlasov et al., 2011). However, since the investigated measurements are not conducted in a field-aligned geometry, our analysis focuses on the electron density to avoid any impacts of ionospheric electric fields (Williams, 1989). Figure 1 shows the electron density $N_e$ measured on 7 July 2020 with the EISCAT VHF radar. A sliding window filter with a 60 s step size and a window length of 60 min is applied on each altitude level separately to subtract all larger scale perturbations

and to disclose the underlying GW signatures. The filtered absolute electron density variations $\delta N_e$ are shown in Fig. 1 as well.

The electron density shows variations of several orders of magnitudes across the observed altitude and time range. The sliding window filter removes the background mean and large-scale variations in time and altitude. The remaining residual fluctuations reveal several medium-scale structures in electron density. Further on, we focus on the pronounced wave structure visible from 8 - 12 UTC at altitudes $\sim 110 - 170$ km indicated by a red box.

Figure 2 (top, left) shows $\delta N_e$ at the time-altitude range specified above. After 10 UTC, the wave structures show signs of an interference pattern below 130 km altitude. This indicates the presence of more than one wave mode, presumably propagating in opposite vertical directions. It should be noted that upward propagating AGW-TIDs have downward phase progression and vice versa (Williams, 1989; Kirchengast, 1997; Vlasov et al., 2011). We apply 2D Fourier filters to separate multiple present wave modes. This approach has been successfully demonstrated on model data (e.g., Vadas and Becker, 2018; Vadas and

Becker, 2019). The electron density measurements are interpolated on a 72 s $\times$ 5 km grid and a 2D Fast-Fourier-Transform (FFT) is performed. The resulting Fourier spectrum is shown in Fig. 2 (top, right) with wave frequency $f$ as abscissa and




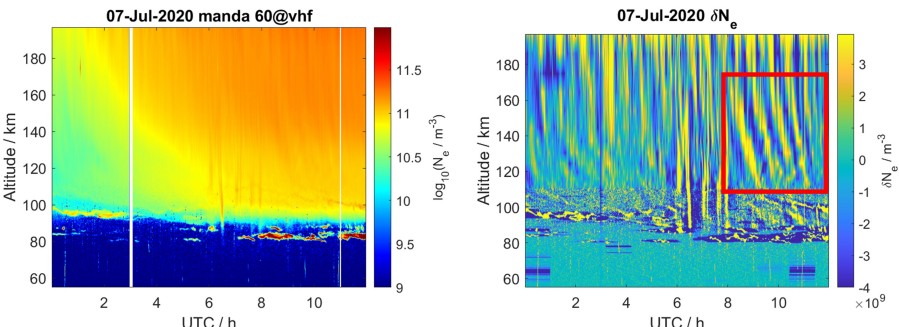

**Figure 1.** Left: electron density measured with the EISCAT VHF radar on 7 July 2020. Right: electron density variation calculated with a sliding window filter. The red box marks a strongly pronounced wave structure.

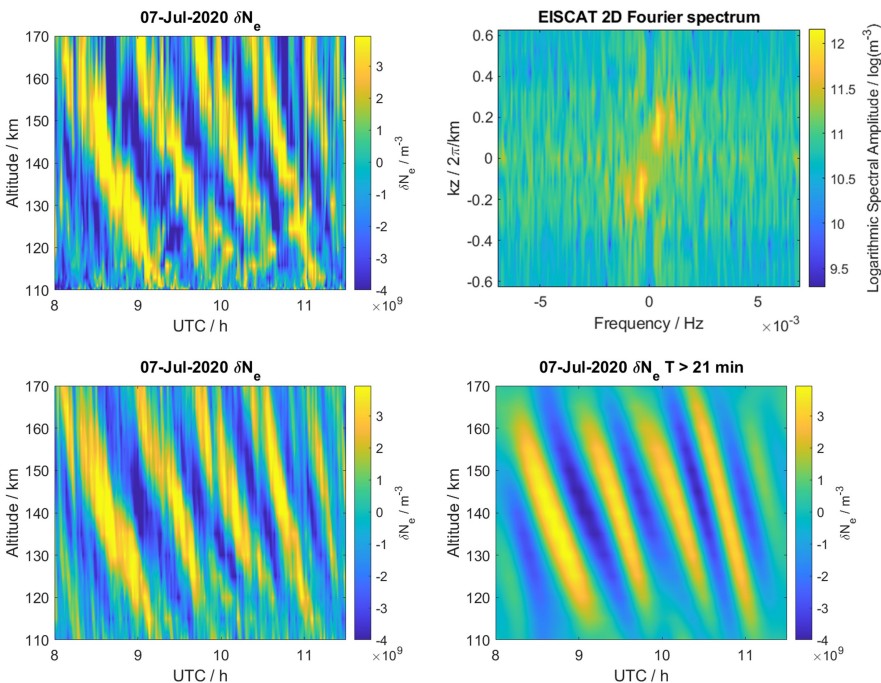

**Figure 2.** Top: electron density variation $\delta N_e$ (left) and associated 2D Fourier spectrum (right). Bottom: $\delta N_e$ filtered for upward propagating wave signals (left) and with additionally restricted wave parameters $\tau \gtrsim 21$ min and $\lambda_z \gtrsim 21$ km (right).

vertical wave number $k_z$ as ordinate. Two strong maxima corresponding to the observed upward-propagating wave structure can be identified in the first and third quadrants of the spectrum. The 2D step function given in Eq. 1 is applied to suppress waves with upward phase progression from the Fourier spectrum.



$$\sigma\left(f, k_z\right) = \begin{cases} 0 & f \cdot k_z \leq 0 \\ 1 & f \cdot k_z > 0 \end{cases} \qquad (1)$$

Figure 2 (bottom, left) shows the filtered 2D inverse FFT that only contains the upward propagating wave mode. This filtered wave field exhibits no more signs of wave interference. However, as we use a bandpass filter, there is a possibility that several upward propagating wave modes are still present. The Fourier spectrum in Fig. 2 (top, right) shows two smaller maxima at slightly higher frequencies ($\sim 1$ mHz) than the dominant maxima. Additionally, the dominating maxima are limited to wave

numbers $k_z \lesssim 0.3$ km$^{-1}$. In the second step, another filter function is applied to limit the Fourier spectrum to waves with period $\tau \gtrsim 21$ min and vertical wavelength $\lambda_z \gtrsim 21$ km. The inverse FFT of this spectrum gives the electron density variations presumably caused by a single AGW-TID wave mode which is shown in Fig. 2 (bottom, right). The same method can be applied to obtain any of the other present wave modes. However, to demonstrate the following procedures, we will focus on this largest-amplitude wave.

After the wave mode of interest has been isolated, the goal is to determine vertical profiles of the wave period $\tau$ and the vertical wavelength $\lambda_z$. The first step is to fit the filtered electron density variations at each altitude level separately as a wave function given by

$$\mathrm{d}N_e = A \cdot \cos\left(\frac{2\pi}{\tau} t + \delta\right). \qquad (2)$$

Equation 2 provides the fit function with the wave parameters amplitude $A$, period $\tau$, and phase shift $\delta$. The optimum param-

eters are determined by a least-square fit which yields the vertical profiles $A(z)$, $\tau(z)$, and $\delta(z)$. Furthermore, we determine the times of the maxima from the wave period and phase shift profiles;

$$t_{max}(z) = -\frac{\delta(z) \cdot \tau(z)}{2\pi} + t_0 + n \cdot \tau(z) \qquad (3)$$

where $n$ is a positive integer and $t_0 = 8$ UTC. Connecting the times of maxima gives the vertical phase lines. Along these phase lines, the vertical wave number $k_z = 2\pi/\lambda_z$ can be determined by

$$k_z(z) = \frac{2\pi}{\tau} \frac{\mathrm{d}t_{max}(z)}{\mathrm{d}z}. \qquad (4)$$

Figure 3 (left) shows the fitted $\delta N_e$ pattern, in which four phase lines are labeled by solid red and dashed black lines. The vertical wavelength $\lambda_z$ along the red phase line and the vertical profile of the wave period $\tau$ are shown in Fig. 3 (right). A similar procedure was applied by Vadas and Nicolls (2009). It should be noted that $\lambda_z < 0$ corresponds to downward phase progression, therefore we show the absolute value $|\lambda_z|$.





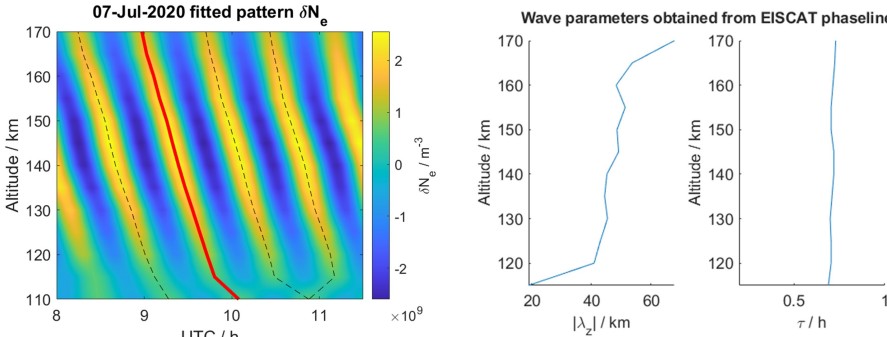

**Figure 3.** Left: fitted wave pattern with phase lines (dashed). Right: profiles of absolute vertical wavelength and wave period for the red phase line.

The obtained profile of the vertical wavelength shows a steady increase from $\sim 20$ km to $\sim 70$ km across the range of measurement altitudes. This agrees very well with previous results from both observations and models (e.g., Oliver et al., 1997; Vadas, 2007; Nicolls et al., 2014). The obtained wave period $\tau = 43.1 \pm 1.6$ min is nearly constant with altitude which also fits previous findings and expectations (e.g., Nicolls et al., 2014).

Though the Fourier filter has to be adjusted manually to account for the specific wave activity during a certain measurement time, the described procedure is an effective method to obtain vertical wave parameters from EISCAT measurements.

### 3.2 Meteor radar

Horizontal wave parameters are derived from measurements with the Nordic Meteor Radar Cluster. Figure 4 (left) shows the total horizontal wind velocity over the Nordic countries at 96 km altitude that was observed on 7 July 2020 at 10 UTC. These horizontally resolved 3D winds are analyzed to extract gravity waves, which then are linked to the TID measured with EISCAT. A time-altitude cross-section of wind measurements at $69°$ N, $22°$ E is shown in Fig. 4 (right).

In the first step, we identify potential gravity waves in the time-altitude domain for each grid cell in the Nordic domain. This is done by filtering for a frequency band around the above-measured TID wave period of $\tau \approx 43$ min. Waves on this time-scale should be resolved in the 10 min resolution meteor radar measurements and be roughly constant with altitude. The analysis of time-altitude dynamics of meteor radar measurements is equivalent to the EISCAT analysis in the previous Section. The main steps are illustrated in Fig. 5 for the selected grid-cell at $69°$ N, $22°$ E.

Due to strong, large-scale changes in the background wind velocity with time and altitude, typical for northern hemispheric summer conditions at such high latitudes, a sliding window filter is applied with 60 min window length and 10 min timesteps. The absolute velocity variations $\delta u$ (top, left) show signs of wave activity above 85 km at $\sim 9 - 11$ UTC. Applying a Fourier filter that allows only upward propagating waves with periods 28 min $\lesssim \tau \lesssim 56$ min, we extract the underlying gravity wave mode at the cost of a slightly decreased amplitude (top, right). Fitting the wave pattern (bottom, left) and determining the wave parameters (bottom, right), as described in Section 3.1, shows that the parameters of the detected wave mode fit well to those




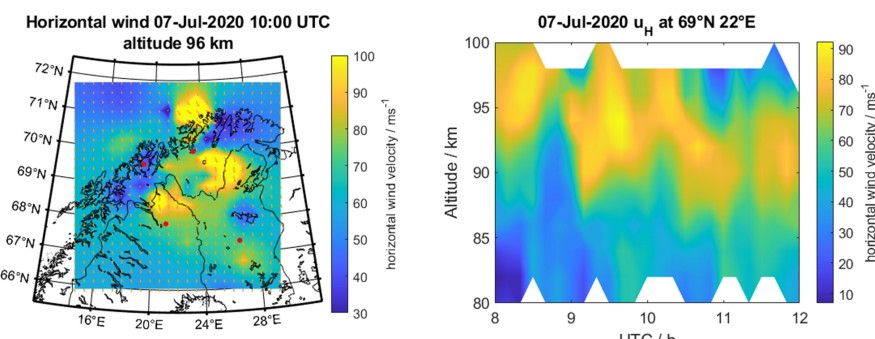

**Figure 4.** Left: Total horizontal wind velocity $u_H$ measured by the Nordic Meteor Radar Cluster at 96 km altitude on 7 July 2020 at 10 UTC, the positions of the four meteor radars are marked as red dots. Right: Time-altitude cross-section of $u_H$ at 69° N, 22° E.

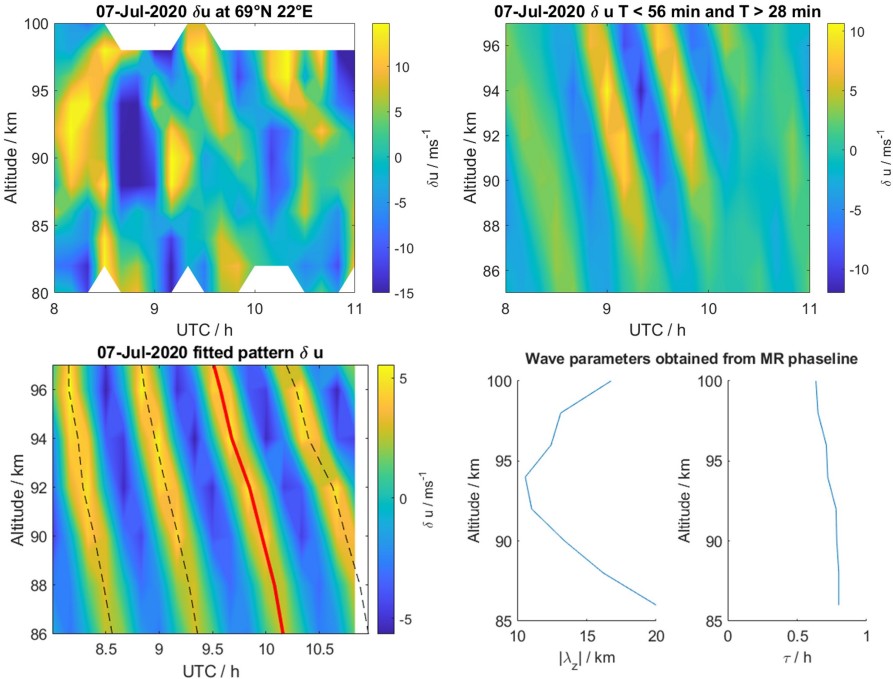

**Figure 5.** Top: Absolute variation of horizontal velocity $\delta u$ in a time-altitude cross-section of the meteor radar measurements at 69° N, 22° E (left). Fourier filtering shows strong wave activity at 28 min $\lesssim \tau \lesssim$ 56 min (right). Bottom: The wave fitting (left) and phase line/wave parameter determination (right) methods are adapted from Section 3.1 and show that the wave parameters of the largest-amplitude wave agree well with the previously detected TID.

found in the EISCAT data at higher altitudes.

The wave period $\tau = 44.1 \pm 4.0$ min is nearly constant with altitude and is within the uncertainties of the wave period measured



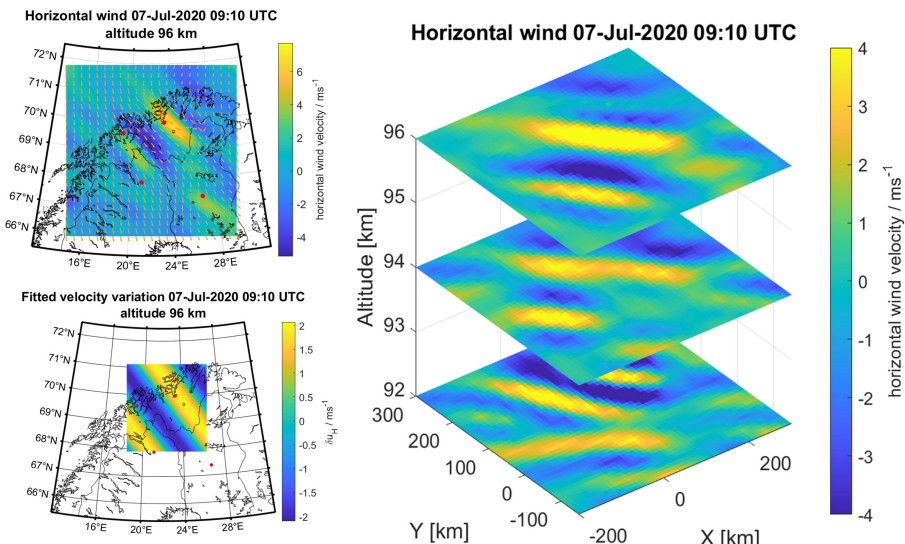

**Figure 6.** Left: North-East-ward propagating wave mode found in the filtered wind field (28 min $\leq \tau \leq$ 56 min) at 96 km altitude (top) and fitted wave pattern (bottom). Right: The wave mode can be detected at multiple altitude layers.

with EISCAT. The vertical wavelength strongly varies with altitude, exhibiting a minimum at approximately the altitude where
the mesospheric summer wind reversal boundary occurs according to existing climatologies involving also some radars of the Nordic Meteor Radar Cluster (Stober et al., 2021d). At $\sim 100$ km altitude, both total value ($\lambda_z \approx 20$ km) and general trend fit well with the lowest altitudes of the profile shown in Fig. 3. This suggests that the detected wave mode is equivalent to the one seen in the EISCAT observations.

The analysis is repeated for each grid point of the Nordic Meteor Radar Cluster to obtain a horizontal field of filtered wind
velocities. The horizontal wind field is Fourier filtered at each altitude level to emphasize the dominant horizontal wave numbers $k_x$ and $k_y$ This reveals a north-east-ward propagating wave mode that is strong enough to be detected at several altitudes. The result of the horizontal wave analysis is shown in Fig. 6.

The horizontal wave parameters of interest are the horizontal wavelength $\lambda_H$ and the propagation direction. Similar to the procedure for the vertical wave fitting, a horizontal wave function is defined in Eq. 5.

$$\mathrm{d}U = A \cdot \sin\left(\frac{2\pi}{\lambda_H} \cdot (\cos\alpha \cdot x + \sin\alpha \cdot y) + \delta\right) \tag{5}$$

The wave propagation direction is defined as an angle $\alpha$ that rotates counter-clockwise from the geographical East direction. The least-square fit includes a phase shift $\delta$ and a horizontally constant amplitude $A$. Figure 6 (top left) shows that the amplitude is indeed not horizontally constant and therefore the fit is conducted on a reduced horizontal area from $\sim 67.8° - 71.1°$ N latitude and $18.4° - 27°$ E longitude. The fit shown in Fig. 6 (bottom left) yields a horizontal wavelength $\lambda_H = 230$ km and
a propagation direction of $\alpha = 36.9°$. Furthermore, Fig. 6 (right panel) outlines the performance of the 3DVAR+DIV retrieval





to infer the vertical and horizontal structure of such gravity waves. Although the sampling is given by randomly occurring meteors in space and time, the algorithm preserves the wave structure for the domain with a sufficient measurement response (measurement response not shown in this work, see for an example Stober et al., 2022).

### 3.3 Dispersion relation fit

The possibility of using AGW-TID observations and gravity wave dispersion relations (Hines, 1960; Vadas and Fritts, 2005) to infer neutral atmosphere parameters has been demonstrated in previous works (see e.g., Nicolls and Heinselman, 2007; Vadas and Nicolls, 2008, 2009; van de Kamp et al., 2014). In particular, obtaining the vertical wave number $k_z$ from ISR measurements has been established in these works. However, the simultaneous measurement of the vertical and horizontal wavelengths of the same wave mode has been difficult due to a lack of the observational capabilities of previous research

instruments. Combining ISR measurements with the Nordic Meteor Radar Cluster provides a unique research capability to measure the thermospheric neutral wind covering the required spatial and temporal scales to enable such studies. The gravity wave dispersion relation gives the wave vector $\mathbf{k}^2 = k_H^2 + k_z^2$ as

$$\mathbf{k}^2 = \frac{N^2 k_H^2}{\omega_I^2} \cdot \gamma - \frac{1}{4H^2} \tag{6}$$

with the Brunt-Väisälä (buoyancy) frequency $N = \sqrt{-g/\rho(z) \cdot \partial \rho(z)/\partial z}$, the atmospheric mass density profile $\rho(z)$, a

viscosity term $\gamma$ and the atmospheric scale height $H$. The intrinsic wave frequency $\omega_I = 2\pi/\tau - \mathbf{k_H} \cdot \mathbf{U_H}$ introduces the effect of the background horizontal wind velocity vector $\mathbf{U_H}$. We can rewrite this by introducing the wind velocity along the propagation direction of the wave $U_\parallel = (\mathbf{k_H} \cdot \mathbf{U_H})/|\mathbf{k_H}|$. The viscosity term of the anelastic dissipative dispersion relation is, according to Vadas and Fritts (2005), given by:

$$\gamma = \left\{ \left[ 1 + \frac{\nu^2}{4\omega_I^2} \left( \mathbf{k}^2 - \frac{1}{4H^2} \right)^2 \left( \frac{1 - Pr^{-1}}{1 + 0.5\delta\left(1 + Pr^{-1}\right)} \right)^2 \right] \cdot \left[ 1 + \delta\left(1 + Pr^{-1}\right) + \delta^2 Pr^{-1} \right] \right\}^{-1} \tag{7}$$

with $\delta = \nu k_z/H\omega_I$, kinematic viscosity $\nu$ and the Prandtl number $Pr = 0.7$. In this study, the Prandtl number is assumed to be constant (Vadas and Fritts, 2005; Nicolls and Heinselman, 2007). In zero-viscosity approximation ($\gamma = 1$), Eq. 6 becomes the dispersion relation derived by Hines (1960). The neutral background atmosphere is taken from the NRLMSISE-00 (Picone et al., 2002) and the kinematic viscosity is calculated from the Sutherland model (Sutherland, 1893). Equation 6 is solved for the optimum wind velocity $U_\parallel$ applying a nonlinear least-squares fit using a Levenberg-Marquardt algorithm (Marquardt,

1963). Figure 7 shows vertical profiles of the wind velocity along the propagation direction of the detected gravity wave. We compare our results from both the viscous and non-viscous dispersion relation, measurements from the Nordic Meteor Radar Cluster projected to the AGW-TID propagation direction, and the empirical Horizontal Wind Model HWM14 (Drob et al., 2015) in Fig. 7.

It can be seen that the fitted wind velocities obtained from viscous and non-viscous dispersion relation agree very well up to

approximately 140 km altitude. Above that, the fit of the non-viscous dispersion relation no longer converges. This is mainly





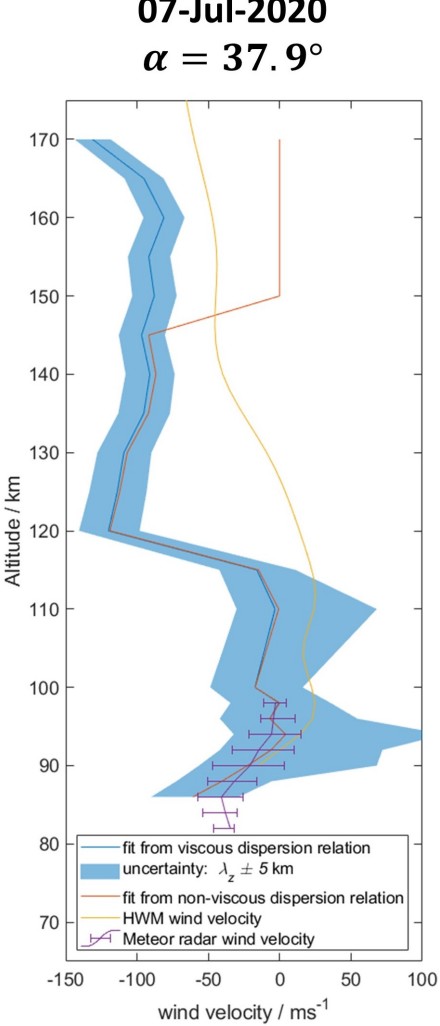

**Figure 7.** Comparison of wind velocities from dispersion relation fit, meteor radar measurements projected to the wave propagation direction, and the HWM14. The shaded area shows the sensitivity of the fit for variations within $\lambda_z \pm 5$ km.

caused by the exponential increase of the kinematic viscosity that results in a breakdown of the zero-viscosity approximation. At altitudes $\lesssim 100$ km, the fitted profiles are well within the range of the projected meteor radar measurements and associated uncertainties. The error bars in Fig. 7 show the upper and lower quartiles of all meteor radar wind velocity measurements during the interval 9 - 11 UTC. The comparison between the winds measured by the meteor radars and those derived from the
wave parameters exhibits a reasonable agreement considering both statistical uncertainties (shaded blue area and error bars). This provides some confidence and validation of the applied approach to ensure that the neutral winds are reliable within the frame of the involved assumptions. The fitted wind velocity profiles follow the general trend of the profile given by the HWM14 though the exact velocities are significantly different. Since the HWM14 is an empirical model aiming to capture only





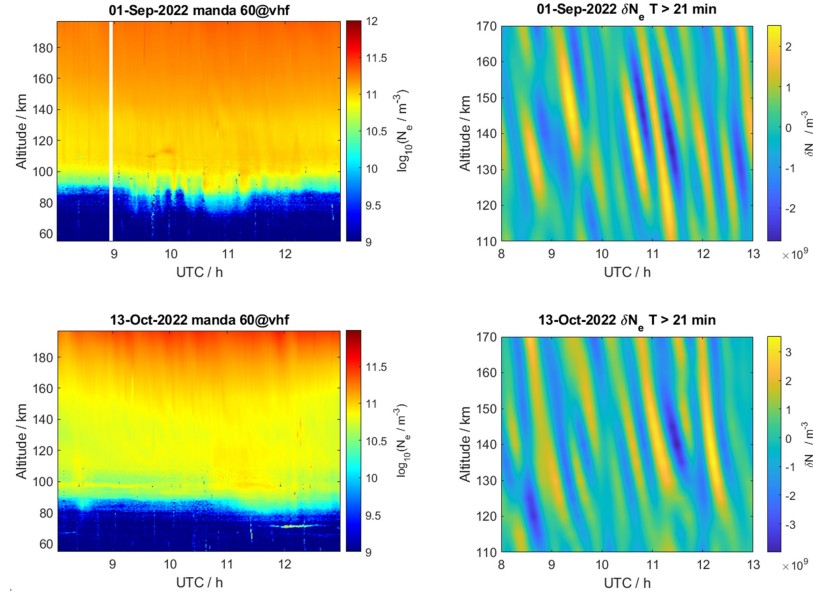

**Figure 8.** Electron densities measured with the EISCAT VHF radar (left) and TIDs filtered for $\tau > 21$ min (right). Shown are the measurements before (1 September, top) and after (13 October, bottom) the fall transition.

the statistical climatology average wind velocity such discrepancies are expected. To emphasize the sensitivity of the velocity
fit procedure, the variations of the result for $\lambda_z \pm 5$ km are shown as the shaded area in Fig. 7. It can be seen that, especially at altitudes below 110 km, variations within a few kilometers of the vertical wavelength can have a quite significant impact on the inferred wind velocity. This indicates that a more accurate determination of the wave parameters is required.

## 4    AGW-TID parameters during the fall transition

Neutral wind observations at the altitude region from $90-150$ km are important to investigate the E-region dynamo and vertical
coupling as well as dynamical coupling processes between the ionized and neutral atmosphere (Baumjohann and Treumann, 1996). The above-described method to derive neutral winds at E-region altitudes leveraging AGW-dispersion relations and multi-instrument observations opens the opportunity to study these processes under various conditions throughout the year. In the following, we apply this method to an AGW-TID event that occurred during the MLT fall transition. The fall transition is connected to the autumn equinox and has been shown to have a major impact on atmospheric tides in the MLT region
(Stober et al., 2021d; Pedatella et al., 2021; Günzkofer et al., 2022). Other studies suggest that there is also an impact on the gravity wave forcing from below (Placke et al., 2015) which, in consequence, will alter the observed wave parameters in the ionosphere. Figure 8 shows two electron density measurements collected with the EISCAT VHF radar on 1st September and 13th October 2022. The data are processed as described in Section 3. Both measurements exhibit signatures of TID activity indicating oscillation periods longer than $\tau > 21$ min, as visualized in Fig. 8 (right panels).



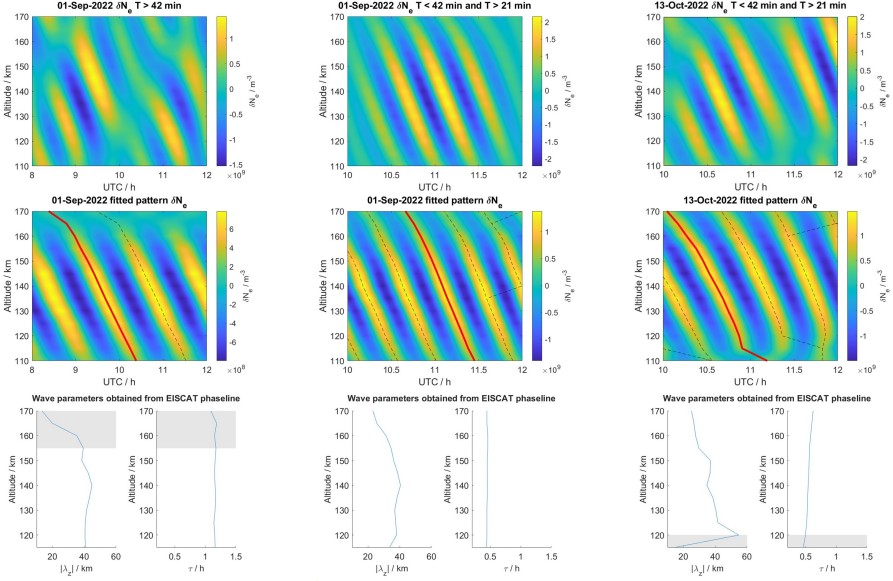

**Figure 9.** Top: EISCAT VHF electron density variations filtered to isolate each of the three identified wave modes. Middle: Fits of the wave modes and the obtained phase lines. Bottom: Wave parameters determined for the red phase lines, the shaded areas indicate altitudes with a range normalized root-mean-square error $NRMSE > 0.25$.

As expected, the electron density is reduced for the October measurement compared to September, due to the generally lower elevation angle of the sun. Furthermore, the filtered data show gravity wave modes at around 10 - 12 UTC with wave periods shorter than one hour. We also want to point out that the measurements on 1st September indicate the presence of a second wave mode with a notably larger period around 8 - 10 UTC. Applying our analysis procedure as described in Section 3, the identified wave modes are separated and the gravity wave parameters are determined. Figure 9 shows the filtered (top)

and fitted (middle) wave modes as well as the determined wave parameters (bottom).

Both TIDs observed on 1st September show a similar profile of the vertical wavelength indicating a gentle increase up to the maximum at $\sim 140$ km altitude with $\lambda_z \approx 40$ km. Above that, the vertical wavelength decreases rapidly. However, the analysis of the longer period TID, detected between 8 - 10 UTC, exhibits an increased uncertainty above 155 km altitude. Both TIDs show a nearly constant wave period in the altitude range from $110 - 170$ km. The long period TID in Fig. 8 has a mean wave

period of $\tau = 69.2 \pm 1.4$ min and for the other TID, we obtained a period of $\tau = 26.9 \pm 0.4$ min. The TID observed on 13th October shows a steady decrease in vertical wavelength from $120 - 170$ km. The sharp increase in wavelength below 120 km is caused by an inaccurate fit of the pattern, as indicated by the increased fit uncertainty, and is therefore not physical. The vertical wavelengths resemble similar values between $\sim 20 - 40$ km in comparison to the TIDs that were found in our first measurements. The mean wave period is $\tau = 33.5 \pm 2.0$ min and shows a slight tendency for a small increase in the period with

increasing altitude. This is also reflected in the increased uncertainty.

It should be noted, that the two TIDs at 10 - 12 UTC, though occurring at the same time of the day, exhibit notably different





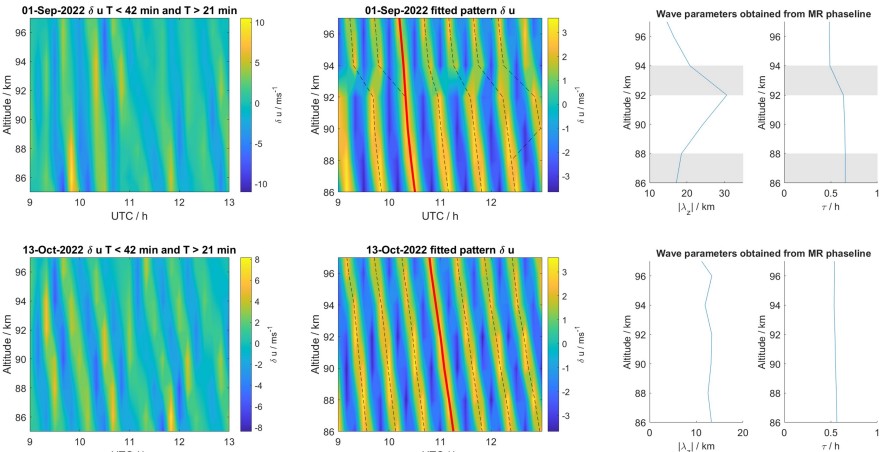

**Figure 10.** Left: Time-altitude cross-section of filtered horizontal wind variations measured with the Nordic Meteor Radar Cluster at $69°$ N, $22°$ E on 1 September (top) and 13 October (bottom) 2022 at 9 - 13 UTC. Middle: Fitted gravity wave oscillations and obtained phase lines. Right: Wave parameters determined for the red phase lines. Shaded areas indicate $NRMSE > 0.35$.

| date | 1 September 2022 | 1 September 2022 | 13 October 2022 |
|---|---|---|---|
| time | 8 - 10 UTC | 10 - 12 UTC | 10 - 12 UTC |
| $\tau$ [min] from EISCAT $(110 - 170$ km) | $69.2 \pm 1.4$ | $26.9 \pm 0.4$ | $33.5 \pm 2.0$ |
| $\tau$ [min] from Meteor Radar $(86 - 98$ km) | - | $32.9 \pm 7.1$ | $32.9 \pm 0.7$ |

**Table 1.** Summary of determined wave periods for the three detected TIDs/AGWs.

wave periods and vertical wavelength profiles. This might be the first indication of the impact of the MLT fall transition on the parameters of AGW-TIDs. The next step is to identify the observed TIDs in measurements of the Nordic Meteor Radar Cluster. As shown in Section 3, the meteor radar data are going to add horizontal information about the gravity waves and also

provide information about the changes in the mean background winds during the fall transition. Since there was no signature in the meteor radar data of an AGW corresponding to the larger-scale TID observed on 1st September, the following analysis is restricted to the two TIDs occurring around 10 - 12 UTC. A time-altitude cross-section of the filtered waves and the parameter analysis is shown in Fig. 10.

The horizontal winds from both measurements were filtered for 21 min $< \tau <$ 42 min. Most notably, below about 92 km,

both AGWs are observed at similar wave periods slightly longer than 30 min, which is closer to the TID wave period found for the October event. While the AGW observed during October shows a constant wave period throughout the entire altitude range, the September AGW shows a transition to shorter wave periods at about $92 - 96$ km altitude. The wave period above this transition is $< 30$ min and close to the period of the September TID. A summary of the wave periods from the ISR and meteor radar AGW-TIDs is given in Table 1.



More important are the altitude-dependent changes of the vertical wavelength for both campaign periods during the fall transition. In September, the AGW exhibits a strong peak in vertical wavelength at 92 km altitude, whereas the October AGW event shows a nearly constant vertical wavelength at all observed altitudes in the meteor radar-derived winds. At about 92 km our fitting method seems to suffer from rather large uncertainties due to a weaker amplitude of the filtered signal. Apparently, at this altitude, other processes disturb the vertical propagation of the AGW. It should be considered that during the fall transition at the beginning of September, the classical circulation pattern changes from the typical summer situation with the mesospheric zonal wind reversal with a strong vertical shear to a weaker mean background wind around October before the winter circulation establishes (Stober et al., 2021d). Such vertical wind shears alter the vertical propagation conditions and, thus, can lead to changes in the observed gravity wave parameters for an observer in an Earth-fixed coordinate frame. Another possible cause for such a strong vertical shear at the MLT is related to atmospheric tides. In particular, the semidiurnal tide exhibits a sudden increase in amplitude during September and shows rather short vertical wavelengths posing favorable conditions to cause strong vertical shears in the flow. Furthermore, the semidiurnal tidal enhancement lasts only a few weeks around the beginning of September and disappears towards October, which further underlines the different dynamical situations of the large-scale flow between the two campaign days during the fall transition. Figure 11 shows the tidal amplitude and phase of the semidiurnal tide over Tromsø from the end of summer (August) until the end of the fall transition in October. The red vertical lines label the campaign days. It is evident that for the event on 1st September, the semidiurnal tide showed a rather short vertical wavelength of about 40-60 km providing favorable conditions to generate a strong vertical wind shear considering also the enhanced amplitude during this period. Hence, the year 2022 is representative of the typical climatological behavior for the fall transition and the evolution of the semidiurnal tidal amplitude and phase (Stober et al., 2021d).

As described in Section 3, the wave period filtering is repeated for all grid points of the Nordic Meteor Radar Cluster and the horizontal wind field is Fourier filtered around the dominant horizontal wave numbers. Both AGWs can be observed in a horizontal cross-section which is visualized in Fig. 12 (September) and Fig. 13 (October).

Most notably, the two AGWs have different horizontal propagation directions. The September AGW propagates in a southwestward direction at an angle $\alpha = 227.7°$ rotated counter-clockwise from the geographical East. The October AGW travels in a northwestward direction at an angle $\alpha = 137.7°$. It can also be seen that their respective maximum amplitudes occur at different positions, though both AGWs are visible around the geographic coordinates of Tromsø. This increases the likelihood that these GWs correspond to the TIDs detected with the EISCAT VHF radar. The horizontal wavelengths are notably different as well, with $\lambda_H = 150$ km for the September and $\lambda_H = 250$ km for the October AGW event. Both AGWs can be observed at multiple altitude levels at or above 94 km altitude and their horizontal wavelengths remain roughly constant at these altitudes. All wave parameters ($\lambda_z$, $\lambda_H$, $\tau$, and $\alpha$) are determined for the two AGW-TIDs that were found in the measurements at 10 - 12 UTC. Leveraging the results of the wave analysis we infer the vertical profile of background neutral wind velocities. The profiles for 1st September and 13th October are shown in Fig. 14. However, due to the very good agreement at the MLT and lower E-region between the viscous and non-viscous results obtained before, we limited the analysis to the viscous dispersion relation here. Similar, to the previous analysis, the fit is compared to the HWM14 model and the meteor radar measurements for the time interval where the AGW-TID was observed.




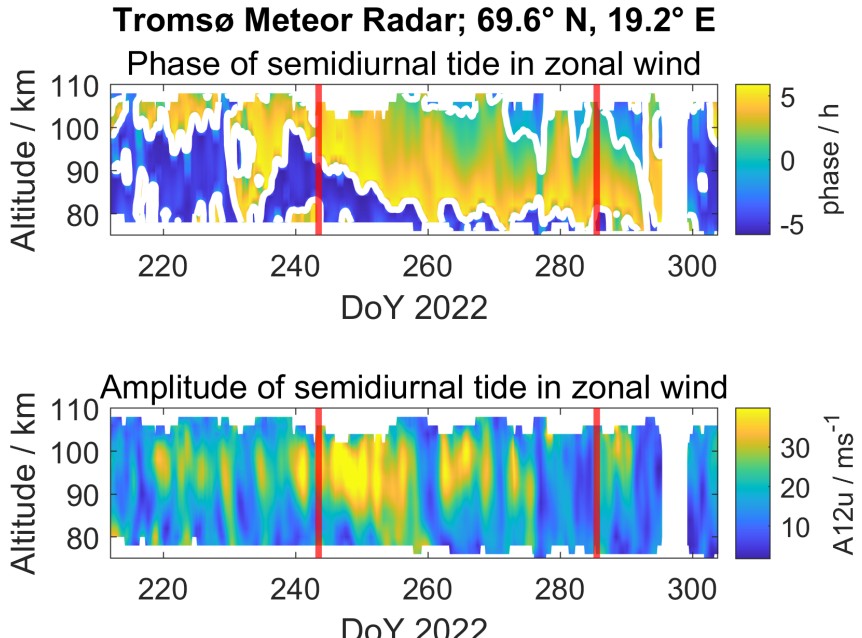

**Figure 11.** Top: Phase of semidiurnal zonal wind tide measured with the Tromsø meteor radar from August till October 2022. The vertical red lines mark the two days of EISCAT measurements. Bottom: The amplitude of semidiurnal zonal wind tide shows a maximum in early- to mid-September.

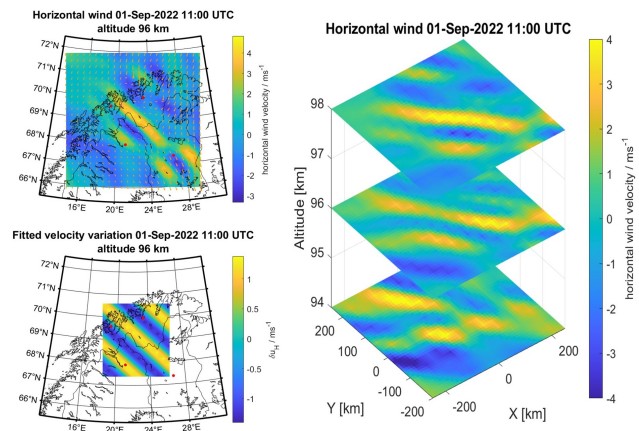

**Figure 12.** Horizontal cross-section of wind variations for 1 September, 11 UTC. Shown are the filtered wind variations (left, top), the wave fit (left, bottom), and a slice plot of three altitude levels (right).

At the altitudes of the meteor radar measurements, both fitted profiles show a similar trend as the measurements, although there are sometimes substantial differences in the absolute values, especially on 1st September around $90 - 94$ km. The largest




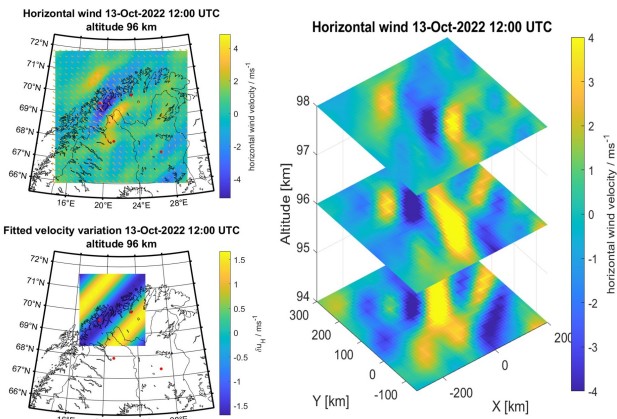

**Figure 13.** Horizontal cross-section of wind variations for 13 October, 12 UTC. Shown are the filtered wind variations (left, top), the wave fit (left, bottom), and a slice plot of three altitude levels (right).

deviations are found for the altitudes of the vertical transition due to the strong vertical shear and corresponding changes in the wind direction and magnitude at approximately $\sim 94$ km that can be seen in Fig. 10. This leads to the conclusion that strong vertical shears imposed by the mean background winds or tides impact the accuracy and precision of the parameter
determination and result in larger uncertainties in the neutral winds derived from the wave parameters. At the altitudes of the EISCAT measurements, the fitted velocity profiles show a similar general trend compared to the HWM14 profiles but sometimes indicate substantial magnitude differences. However, as mentioned in Section 3, this is attributed to the climatology nature of the HWM14 velocities, which cannot reflect specific synoptic situations due to a particular wave field or energetic forcing. The resemblance of the fitted profiles with both measured and modeled profiles is a promising first result for this
method.

## 5   Discussion

Combining observations with the EISCAT radar and the Nordic Meteor Radar Cluster provides a new capability to study AGW-TIDs. The presented approach avoids several problems arising from previous techniques using either multi-beam ISR measurements or a combination of classical ISR and GNSS networks. Measurements with a phased array ISR would in addition
permit the determination of both vertical and horizontal properties of a single wave mode. However, the horizontal resolution of such measurements is limited. Consequently, horizontal wavelength and propagation direction can only be roughly determined (Nicolls and Heinselman, 2007; Vadas and Nicolls, 2008). This should also limit the capability of inferring background neutral winds significantly. On the other hand, GNSS networks allow to measure MS-TIDs with high spatial and time resolution (Saito et al., 1998; Tsugawa et al., 2007). The disadvantage of this technique is that 2D TEC maps do not allow for the separation of
different wave modes which makes a combination with ISR measurements difficult (van de Kamp et al., 2014).

The AGW-TID wave parameters determined in this paper are all within the parameter range ($\lambda_z \sim 10-100$ km, $\lambda_H \sim 100-300$





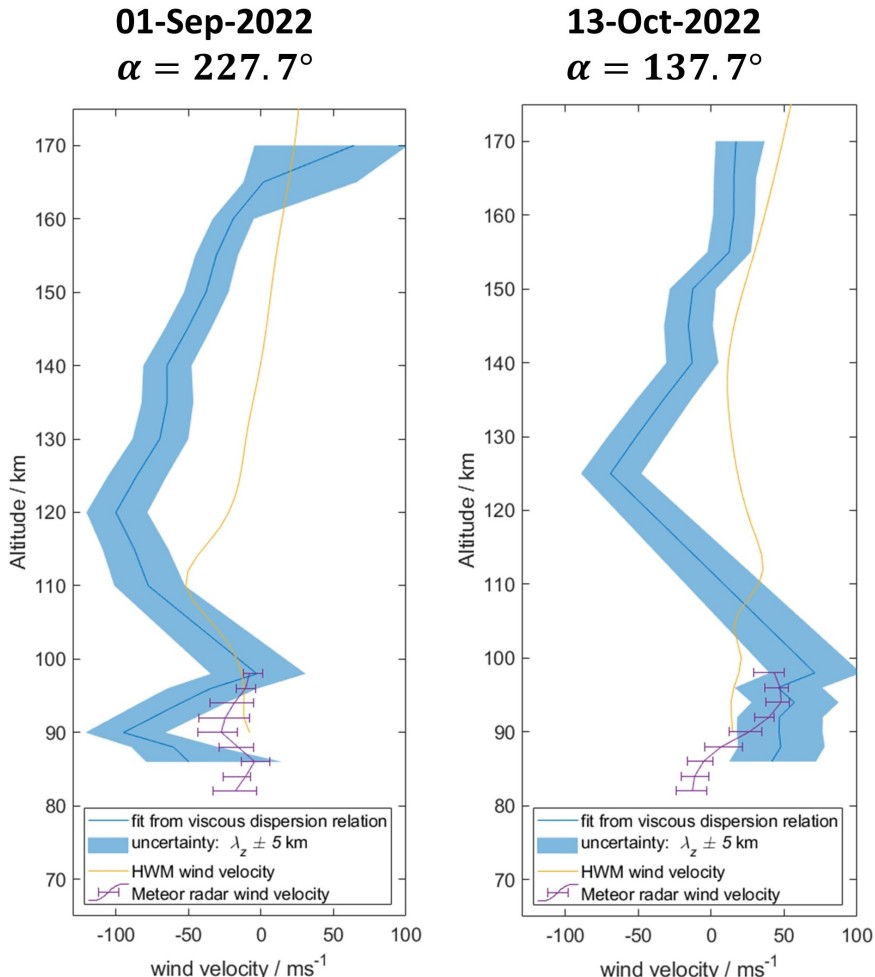

**Figure 14.** Wind velocity profiles along the propagation direction of the AGW-TIDs detected for each of the measurement days. The propagation direction is given as angle $\alpha$ rotated counter-clockwise from the geographic East. The shaded area shows the fit sensitivity for variations $\lambda_z \pm 5$ km.

km, $\tau \sim 20 - 100$ min) found in several previous studies (see e.g., Oliver et al., 1997; Kotake et al., 2007; Nicolls and Heinselman, 2007; Vadas, 2007; van de Kamp et al., 2014; Nicolls et al., 2014). It has been shown that daytime MS-TIDs are mostly connected to an upward propagating AGW from lower atmospheric layers whereas nighttime MS-TIDs can be generated by

electrodynamic processes in the ionosphere, e.g., Joule heating or the Perkins instability (Tsugawa et al., 2007). These in situ generated AGW-TIDs are unlikely to propagate down to the altitudes covered by the Nordic Meteor Radar Cluster. Therefore, the simultaneous detection of daytime AGW-TIDs with the EISCAT radar and the Nordic Meteor Radar Cluster in this study underlines many of the results obtained in previous studies.

Future work should target the investigation of wave parameter changes caused by other atmospheric events besides the MLT



fall transition. Events of special interest might be Sudden Stratospheric Warmings, the Hiccup of the autumn transition, and the spring transition which all show distinct similarities and differences (Matthias et al., 2015; Matthias et al., 2021). Determining wave parameters from simultaneous measurements with additional instruments would help to further refine the demonstrated method and possibly expand the range of investigated altitudes. These could include the well-established MS-TID measurements with GNSS networks as discussed above. Since both the Nordic Meteor Radar Cluster and GNSS networks allow the

determination of the propagation direction, single wave modes might be observed simultaneously and thereby linked to EISCAT measurements. Statistical studies of daytime and nighttime MS-TIDs suggested that the preferred propagation direction of these waves depends on the generation mechanism (Kotake et al., 2007; Tsugawa et al., 2007). These studies were conducted on measurements in the North American region which makes a comparison to our measurements in Fennoscandia difficult. Based on the works of van de Kamp et al. (2014), such studies could be conducted in this region combined with EISCAT and

Nordic Meteor Radar Cluster measurements. The application of OH airglow spectrometers has been previously demonstrated (Wüst et al., 2018; Sarkhel et al., 2022) and could be applied as well. There are several planned satellite missions targeting the detection of AGWs in the MLT region that might provide valuable additional information (e.g., Sarris et al., 2020; Gumbel et al., 2020). Comparison to a gravity wave resolving atmosphere model like the High Altitude Mechanistic General Circulation Model (HIAMCM) (Becker and Vadas, 2020) would give valuable insight into the origin and generation of observed waves.

This includes the potential role of secondary and tertiary gravity waves generated in the mesosphere and thermosphere (Vadas and Becker, 2018; Vadas and Becker, 2019) and the polar night jet (Becker et al., 2022b, a; Vadas et al., 2023). Validation of the inferred velocity profiles above 100 km altitude is difficult with the presently available instruments. However, the EISCAT3D system (Stamm et al., 2021) could enable altitude-resolved multi-static ISR measurements from which neutral winds could be inferred.

## 6 Conclusions

It has been shown that vertical and horizontal wave parameters of AGW-TIDs can be determined from simultaneous measurements with the EISCAT VHF radar and the Nordic Meteor Radar Cluster. Such observations allow studying the vertical coupling processes and propagation of AGW-TIDs. EISCAT and meteor radar measurements can be combined since they are only separated by about $10 - 20$ km in altitude. High time resolution multi-static meteor radar measurements at 10 min steps

allow us to estimate the wave period and therefore specifically filter out wave modes detected in EISCAT measurements. The developed techniques to filter wave modes and determine wave parameters can be adapted to other EISCAT and meteor radar campaigns. We demonstrated the application of this method on two measurement campaigns conducted in early September and mid-October 2022, before and after the MLT fall transition. In both measurements, an AGW-TID occurring around 10 - 12 UTC with a wave period of roughly 30 min were detected. We showed that both waves exhibited a similar parameter range

below $\sim 90$ km. The September AGW-TID underwent notable changes in vertical wave parameters that were detected in the ionosphere. This shows that the fall transition impacts the ionospheric variability due to the amplification of semidiurnal tides in early September and the tidal minimum in October. Our study also shows that it is possible to apply the determined wave



parameters to infer neutral wind velocity profiles in the thermosphere. While the absolute values of the inferred, measured, and modeled wind velocities did not always agree, the general trend of the profiles showed remarkable agreement considering the typical statistical errors. This indicates that this method provides a possibility for reliable neutral wind estimates in the thermosphere, given more refinement and validation. A more extensive data collection from the multi-instrument AGW-TID measurements discussed above, including ISR, meteor radar, GNSS, ground- and satellite-based airglow imagers as well as explicit wave simulations is going to improve the database to study the vertical coupling and permit further refinement of the applied procedures. Extending these studies to other events, e.g. Sudden Stratospheric Warmings, will help understand the impact of atmospheric variability on the ionosphere.

*Data availability.* The data are available under the Creative Commons Attribution 4.0 International license at https://doi.org/10.5281/zenodo.7752777 (Günzkofer et al., 2023). Please contact Alexander Kozlovsky (alexander.kozlovsky@oulu.fi) for the Nordic Meteor Radar Cluster 3DVAR+DIV retrievals.

*Author contributions.* FG performed the data analysis and wrote large parts of the manuscript. DP, GS, and IM suggested the idea for the multi-static EISCAT experiment and were the PIs of the July 2020 EISCAT campaign. AT provided the analysis of ion velocity vectors and helped to plan the EISCAT experiments. SV suggested the application of Fourier filters and SV and EB provided feedback on AGW-TID analysis. AK, MT, NG, SN, ML, EB, JK, and NM are PIs of the Nordic Meteor Radar Cluster. All authors provided feedback and were involved in revising the manuscript. The supervision of FG is supported by the University Bern.

*Competing interests.* Three of the (co-)authors are members of the editorial board of Annales Geophysicae.

*Acknowledgements.* EISCAT is an international association supported by research organizations in China (CRIRP), Finland (SA), Japan (NIPR), Norway (NFR), Sweden (VR), and the United Kingdom (UKRI). This work uses pyglow, a Python package that wraps several upper atmosphere climatological models. The pyglow package is open-sourced and available at https://github.com/timduly4/pyglow/. This research has been supported by the STFC (grant no. ST/S000429/1), and the Japan Society for the Promotion of Science (JSPS, Grants-in-Aid for Scientific Research, grant no. 17H02968). This research has been supported by the Schweizerischer Nationalfonds zur Förderung der Wissenschaftlichen Forschung (grant no. 200021-200517/1). The Esrange meteor radar operation, maintenance, and data collection were provided by the Esrange Space Center of the Swedish Space Corporation. The Nordic Meteor Radar Cluster data analysis and calculations were performed on UBELIX (http://www.id.unibe.ch/hpc, last access: 20 February 2023), the HPC cluster at the University of Bern. Gunter Stober is a member of the Oeschger Center for Climate Change Research (OCCR). Ingrid Mann is supported by the Research Council of Norway, NFR 275503.



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
