# Peer review of "Inferring neutral winds in the ionospheric transition region from AGW-TID observations with the EISCAT VHF radar and the Nordic Meteor Radar Cluster"

_EGUsphere, 2023_

## Author Comment (AC1)

**2D Heaviside filter**

2D Fourier spectrum filtered by propagation direction

wind fields from retransformed filtered Fourier spectra

example wind field from two interfering modes ("checkers pattern")

2D Fourier spectrum

**2D FFT**

**2D IFFT**

---

## Author Response (AR1)

**Response to Referee comments**

**Referee 1:**

The manuscript presents more advanced and refined methods compared to previous ones for detecting and analysing atmospheric gravity waves and traveling ionospheric disturbances with incoherent scatter and meteor radars. Both instruments seem to detect the same AGW-TID activity on the in total three days used for the study. AGW parameters are found to be different depending on the seasonal phase relative to a "fall transition". The manuscript is relatively well written, but sometimes descriptions seem to be inaccurate and equations incomplete. The figures are of good quality. Before a publication I suggest a few improvements according to my comments below.

We thank the referee for their feedback.

Referencing:

Our friends from the LOFAR community have in the recent years been successfully studying AGWs at small to medium scales, which should perhaps be mentioned together with other methods refered to in lines 45-55. I found these papers

Boyde et al. (2022), Lensing from small-scale travelling ionospheric disturbances observed using LOFAR, https://www.swsc-journal.org/articles/swsc/full_html/2022/01/swsc220042/swsc220042.html

Fallows et al. (2020), A LOFAR observation of ionospheric scintillation from two simultaneous travelling ionospheric disturbances, https://www.swsc-journal.org/articles/swsc/full_html/2020/01/swsc190078/swsc190078.html

and a presentation at the EGU 2023: https://meetingorganizer.copernicus.org/EGU23/session/46346#Orals

Though not covering the same geographic locations and so far also not the same time periods, a general comparison between the LOFAR and here presented EISCAT methods and their advantages could be briefly attempted, e.g. in Section 5.

The LOFAR studies have been added to the list of previous measurements in the introduction and the capabilities and results are briefly discussed in comparison to EISCAT in Section 5.

Lines 15-16: "In the ionosphere, AGWs can be observed as Medium-Scale Traveling Ionospheric Disturbances (MS-TIDs) from neutral-ion collisions (Nicolls et al., 2014)." The statement could be clearer. I think that something like --> "In the ionosphere, which is coupled to the neutral atmosphere by ion-neutral collisions, AGWs can be observed as Medium-Scale Traveling Ionospheric Disturbances (MS-TIDs) (Nicolls et al., 2014)."

Changed as suggested.

Lines 135-147: A relation between an upward or downward propagation of the AGW and the occurrence of negative vertical wavenumbers/frequencies seems to be implied, but why and how should be explained in more detail to the reader. In the 3rd quadrant both $k_z$ and $f$ are negative, so their product is positive. The step function, Eq 1, does not

remove any of the strong peaks in the 1st and 3rd quadrants, only the downward propagating "noise" in the 2nd and 4th quadrants. Is this correct?

That is correct. Downward phase progression is usually an indication that the AGW-TID wave is propagating upward. Strong background wind shear could reverse the phase behavior; however, this is presumably not the case for the shown measurements.

The fact that downward phase progression corresponds to | f * k_z | > 0 in a 2D Fourier spectrum is a mathematical property of the Fourier transform which can easily be seen from an artificial example wave pattern (see figure below):

[Figure]

The explanation has been extended and clarified.

Equation (1) is incomplete as it stands alone. What is sigma and how is the filter exactly working? The reader can guess this relatively easily, but it would be better to have the complete equations of the 2-D Fourier (inverse) transform and the filter.

As suggested, Eq. 1 has been rewritten to a full equation showing how the filter function is applied to the spectrum to give a filtered Fourier spectrum.

Lines 172-173: "The obtained wave period τ = 43.1 ± 1.6 min is nearly constant with altitude which also fits previous findings and expectations (e.g., Nicolls et al., 2014)." Wouldn't be a discussion of the Brunt-Väisälä period be appropriate at this point. I think that the Brunt-Väisälä frequency is not very constant in the altitude range plotted in Figure 3 (because of the transition from molecular to atomic particles (e.g. $O_2$ -> O) and the temperature gradient). The vertical wavelength could be compared with the scale height of a hydrostatic equilibrium.

The Brunt-Väisälä frequency is not constant at these altitudes. However, for k_H << k_z (which is definitely the case here), the observed wave frequency can be much smaller than the buoyancy frequency (see Fritts and Alexander 2003 and Nicolls et al. 2014). A brief summary/discussion has been added to our manuscript.

Line 185: "... for the selected grid-cell at 69◦ N, 22◦ E." Why was this grid cell selected? The EISCAT VHF beam would be at 69.58° N and 19.23° E, where also one of the four meteor radars is located.

In Figure 6 (left, top) you can see that the wave amplitude maximizes towards the east of the EISCAT location. The same wave mode can also be detected at 69.5° N 19.5°E but with a reduced amplitude and therefore we decided to show the vertical cross section of the wave at a position of strong wave amplitude rather than at the exact EISCAT location.

Lines 193-194: "The wave period τ = 44.1±4.0 min is nearly constant with altitude and is within the uncertainties of the wave period measured with EISCAT." Again, I'm a bit skeptical that this period comes out artificially because of the filtering method while physically relevant periods, for example Brunt-Väisälä, have different values and are not so constant over the altitude. Please discuss this.

We refer to Vadas and Becker, 2018 especially Eq. 8, where it can be seen that only the change in time of the background wind and intrinsic wave frequency can change the observed wave period (see Vadas and Fritts, 2005). So, if the buoyancy frequency changes in altitude but not in time (which is usually the case over time periods of a few hours or less), then the observed wave period will be constant in height. The explanation in the paper has been clarified.

Line 200: "The horizontal wind field is Fourier filtered at each altitude level ...", but then in lines 209-210 "The fit shown in Fig. 6 (bottom left) yields a horizontal wavelength λH = 230 km and a propagation direction of α = 36.9◦ ." single, not altitude dependent values are obtained. I thought you would do the whole filtering and fitting at each altitude level and then come up with z-dependent λH and α.

The fitting at several altitude levels is only to confirm that lambda_H and alpha are approximately constant with altitude. This has to be assumed up to the EISCAT altitudes. Since the assumptions could be confirmed for the meteor radar altitudes, constant values are applied for the dispersion relation fit throughout all altitudes. This has been clarified in the text.

Lines 226-227: "We can rewrite this by introducing the wind velocity along the propagation direction of the wave U∥ = (kH · UH ) / |kH |." With this definition, U∥ is a scalar, not a vector with a direction. It is a wind "speed". The subscript ∥ normally refers to the magnetic field direction, but here this is not meant. Rather it is a horizontal direction. The AGW has horizontal velocity and wave (or k) vectors. Both have different directions, and I think that you want to project the wind onto the wave vector direction. Please describe more precisely (also the equation).

U_parallel is the neutral wind projected along the direction of the horizontal wave vector. The explanation has been clarified.

Lines 233-235: "Equation 6 is solved for the optimum wind velocity U∥ applying a nonlinear least-squares fit using a Levenberg-Marquardt algorithm." Again I think that this is incomplete and a bit confusing. Equation 6 has no parameter U∥ that could be optimized. Please describe more comprehensively which expression is minimized with LM, which parameters are observed or are filtered observations, which parameters are from a model, and whether parameters are z dependent or assumed to be constant.

Eq. 6 has been extended to show the parameter U_parallel and the explanation has been extended.

Line 240: "... approximately 140 km altitude. Above that, the fit of the non-viscous dispersion relation no longer converges." but Figure 7 shows a curve for the fit of the non-viscous dispersion relation also above 140 km. The curve seems to be at exactly 0 m/s which probably indicates no convergence. It would be better to not plot the curve at altitudes where the LM did not converge.

The plot has been modified according to your suggestion.

Line 597: "... https://doi.org/110.1029/2017JD027970 ..." --> "... https://doi.org/10.1029/2017JD027970 ..."

Changed.

References:

Fritts and Alexander, 2003,

Nicolls et al., 2014,

Vadas and Becker, 2018,

**Referee 2:**

We thank the Referee for taking the time to read and evaluate our paper and for the kind and helpful review.

**Minor comments**

– In the introduction, I think it could be valuable to highlight a bit more how challenging it is to obtain measurements of the lower thermosphere–ionosphere (LTI) region, where the couplings from above (space) and below (underlying atmosphere) are crucial and still relatively poorly understood. A recent paper reviewed those challenges and discussed in particular the role of atmospheric waves in the LTI dynamics (Palmroth et al., 2021, https://doi.org/10.5194/angeo-39-189-2021); it could be worth referring to it in the introduction.

The difficulty of obtaining measurements in the LTI region is one of the main points of this paper. We added a paragraph to highlight this even more, including a reference to the suggested publication.

– l. 145: If I understand correctly, filtering out the second and fourth quadrants in the FFT panel removes the downward propagating / upward phase wave which is responsible for the interference pattern visible between ~10 and 11 UTC in the top-left panel of Fig. 2. I however am not sure whether I am able to identify the FFT signal corresponding to this wave in the top-right panel. Could it be the slightly brighter spots around ±(0.05, –0.3)? This is just pure curiosity, as I am a bit intrigued by the fact that the signature of this wave mode in the FFT panel does not stand out much.

Yes, it can be assumed that the two light maxima at (+0.05,-0.3) and (-0.05,+0.3) correspond to the downward propagating wave responsible for the interference pattern at about 10-11 UT. Since this wave mode is only present for a short time and only at certain

altitudes, the decreased amplitude of the Fourier maxima compared to the large upward-propagating wave mode seems reasonable.

– l. 150–151: To ease the understanding of the reader, I would suggest adding annotations to the top-right panel of Fig. 2 to show which parts of the FFT are retained at each step. For instance, adding grey shading to the second and fourth quadrants as well as thin lines indicating the cutting wavelengths and frequencies could enhance the figure.

We added black and red rectangles to illustrate the edges of the filters applied to obtain the plots in the bottom row. An explanation has been added to the Figure caption.

– Fig. 4: I am a bit intrigued by the fact that the wind data appear as a square on top of a map with a different type of projection (left panel). Could the Authors briefly comment on this? Besides, it is difficult to distinguish the small orange dots (or are these arrows indicating the horizontal wind direction?) from the background colours. I would recommend using a different colour for these, to enhance the contrast, and perhaps make them longer if these are indeed arrows. Finally, I suggest indicating on the map the location of the point from which the time-altitude cross-section of the vertical wind is shown in the right panel.

The data pre-processing of the Nordic Meteor Radar Cluster aims to provide data on geometrically rectangular grid of 330 x 330 km with 30 km resolution. The projection of the geographical map is chosen specifically to emphasize the rectangular character of the data grid.

The orange "dots" are indeed arrows. We kept the color of the arrows the same due to the guidelines regarding the accessibility of figures for people with color vision deficiencies.

We added a red star to indicate the position of the vertical profile.

Please note that we also changed the time of the plot from 10 UT to 9:10 UT to be identical with the wave shown in Fig. 6.

– l. 182: "This is done by filtering for a frequency band...". Although there is little ambiguity given the context, it could be worth clarifying what parameter is being filtered.

Changed to "This is done by filtering the neutral wind measurements for a…"

– l. 183: "... and be constant with altitude". Probably being a bit too nitpicky here, but I would suggest rephrasing into something like "and their period/parameters should be constant with altitude" (as I am not sure to what extent it is proper to talk about "constant waves").

That's a valid point. The sentence has been clarified in a similar way as you suggested.

– Fig. 5: Is there a particular reason for restricting the shown altitude range in the top-right and bottom-left panels? Since the bottom-right one shows the entire altitude range (85–100 km), I would recommend showing the intermediate-step data across the same range as well.

Due to the poor data coverage above about 98 km (visible in the top left plot), the following plots have been restricted in altitude. The range from 80-85 km has been cut off

since there is no detected wave oscillation at these altitudes. The parameter profiles are shown from 85-100 km so it doesn't seem like we try to cover any changes in the profile trend that would then not agree so well with the EISCAT parameter profile.

– Fig. 6: I am not sure I understand fully the top-left panel. Do the arrows indicate the filtered horizontal wind direction? If so, when the values are negative, does it mean that the wind is opposite the direction of the local arrows? Please add a clarification in the figure caption.

The arrows indicate the total horizontal wind velocity and are therefore the same as in Fig. 4. A note for clarification has been added.

– Fig. 9: I would suggest increasing the font size of the labels in this figure, as they are a tiny bit small to read when printed on A4.

The Figure has been rearranged to show the labels in a readable size.

– Fig. 10: I am not sure I understand how the phase lines shown with dashed lines were determined in the top-middle panel. Some wave front parts seem to belong to two phase lines, and I do not understand how the continuity was chosen at the "jumps" between 92 and 94 km altitude (see in particular the latest phase line which exhibits two jumps and looks a bit strange to me). Please add a brief clarification.

Since the measurements were filtered to include only upward propagating waves (downward phase lines), the algorithm to automatically determine the phase lines only allows downward phase lines. At 92-94 km altitude, some of the wave fronts appear to have upward phase propagation and therefore the phase lines "jump" to the next one in time. The last one shows two jumps simply because there is no data after 13 UT and therefore the phase line determination is forced to jump back to the wave front before.

The parameter determination was specifically done for a wave front that shows continuous downward phase for all altitudes. A short clarification has been added to the manuscript.

– Fig. 12 & 13: Here too, either the figures should be enlarged or the labels should have a larger font size, to improve legibility. The arrow vectors on the top-left panels are also very difficult to distinguish (even with 200% zoom); a different colour could enhance the contrast with the background.

The figures have been rearranged same as Fig 6 and were also enlarged. Color changes are difficult due to the previously mentioned guidelines and have therefore not be done.

– l. 370: The satellite mission described in Sarris et al. (2020), called Daedalus, was unfortunately not selected for phase A. Hence, I suggest slightly rephrasing into "There have been several planned satellite missions...".

As I understand there is another attempt for a mission similar to Daedalus, described in Sarris et al. (2023). The reference has therefore been changed to this new paper.

– l. 377: "EISCAT3D" --> "EISCAT_3D". I think the new method presented in this paper will likely lead to terrific results when EISCAT_3D starts operating, so I would suggest elaborating a little bit more on the applicability of the method and possible further developments with this upcoming new observational capability. In particular, this could

be a great opportunity to bridge the gap between the attmospheric sciences and space physics communities, thanks to the strong multidisciplinarity component of the research questions and methods presented here. A mention to EISCAT_3D could also be made towards the end of the conclusions, if the Authors find it relevant.

We elaborated the use EISCAT_3D to verify inferred profiles and thereby the applied method. This might allow to use the presented method on all possible AGW-TID measurements and lead to a new tool for neutral wind measurements in the lower ionosphere. This has also been added at the end of the conclusion.

– l. 388–389: "An AGW-TID (...) were detected" --> mismatch between singular subject and plural verb.

Changed.

**Additional comments Referee 2:**

I thank the Authors for considering my comments and suggestions.

There is only one aspect on which I would like to insist a bit more. I am very much aware of the guidelines regarding figure accessibility to people with vision deficiency, and I appreciated the fact that the Authors selected colour-blind-friendly colour maps in their figures – I would most definitely have commented on it if it had not been the case. However, the arrows overlaid on horizontal wind maps in Figs 4, 6, 12 and 13 are drawn with a shade present in the colour map used for the background data. This is a suboptimal choice, as some of the arrows cannot be distinguished from the background (no matter the vision capability of the reader), especially in areas where the wave fronts are prominent. I would therefore recommend that the Authors do consider finding a suitable colour for these arrows to improve the legibility of those figure panels. I can suggest for instance trying out various shades of grey or shades of blue darker than the lower end of the colour map. A useful (and free) online tool to assess the compatibility of colour choices with various types of vision deficiencies is Coblis (https://www.color-blindness.com/coblis-color-blindness-simulator/), which the Authors might already be familiar with. If nothing works, then so be it, but I think it would be important to first envisage possible solutions.

Looking forward to the revised manuscript!

We are glad we could address most of the referee's comments. Regarding the color of the arrows in Fig. 4, 6, 12 and 13, we changed the arrow color to a dark grey color. We hope we could improve the figure quality with this. We thank the referee also for pointing out the Coblis tool.

---

## Referee Report (RR1)

**Review of paper #egusphere-2023-678 – "Inferring neutral winds in the ionospheric transition region from AGW-TID observations with the EISCAT VHF radar and the Nordic Meteor Radar Cluster" by Günzkofer et al. (first revision)**

The Authors have addressed all my comments adequately, and I can say in particular that the revised figures look great. The paper reads very well despite the density of information it contains, and I think this is overall a very nice piece of science which opens new possibilities for the study of atmosphere–ionosphere couplings in future.

Please note the technical comments below, which can certainly be addressed at typesetting to not delay the publication process unnecessarily. I am pleased to recommend the manuscript for publication.

**Technical comments**

– Since the layout of Fig. 6 was changed during the revision, the figure caption should be updated accordingly! Same for mentions such as "top left", etc., in the text on l. 220, 222 and 225.

– Same thing for Figs 12 & 13.

---

## Author Response (AR2)

The minor changes suggested by the two referees have been implemented in the manuscript.

Referee 1:

The quoted sentence has been rephrased to be clearer and more readable. The section of the Fritts and Alexander review that contains the important information has been added to the reference.

Referee 2:

The Figure references have been corrected, both in the captions and in the text.

We thank both referees for pointing out these problems.